# Connectedness of loss landscapes via the lens of Morse theory

**Danil Akhtiamov**                                                    DAKHTIAM@CALTECH.EDU
*Computing and Mathematical Sciences*
*Caltech*
*Pasadena, CA 91125*

**Matt Thomson**                                                      MTHOMSON@CALTECH.EDU
*Biology and Biological Engineering*
*Caltech*
*Pasadena, CA 91125*

**Editors:** Sophia Sanborn, Christian Shewmake, Simone Azeglio, Arianna Di Bernardo, Nina Miolane

## Abstract

Mode connectivity is a recently discovered property of neural networks stating that two weight configurations of small loss can usually be connected by a path of small loss. The mode connectivity property is interesting practically as it has applications to design of optimizers with better generalization properties and various other applied topics as well as theoretically as it suggests that loss landscapes of deep networks have very nice properties even though they are known to be highly non-convex. The goal of this work is to study connectedness of loss landscapes via the lens of Morse theory. A brief introduction to Morse theory is provided.

**Keywords:** Mode connectivity, Morse Theory, Loss Landscapes, Saddle Points, Non-Convex Optimization

## 1. Introduction

Many lines of research suggest that geometric and topological properties of the parameter manifold of a neural network play a central role in machine learning. Indeed, arguably the most important algorithm used for training neural networks- stochastic gradient descent- can be interpreted as the dynamical system defined by the gradient flow of the loss function on the weight manifold. These flows define potential solutions to a machine learning problem and thus ultimate parameters for the trained network. Curvature plays an important role in generalization and mode connectivity (see Hochreiter and Schmidhuber (1994), Hochreiter and Schmidhuber (1997), Raghavan and Thomson (2020)). However, formal mathematical tools for studying these geometric properties are still being developed.

One of the most powerful tools in differential topology- Morse theory- studies differentiable manifolds in terms of real-valued functions on them Milnor (2016). This theory provides a way of reconstructing a manifold $M$ given access to the knowledge of critical points of a function $f : M \to \mathbb{R}$ and indexes of Hessians $Hess_x(f)$ at these critical points only (here we mean the number of negative eigenvalues by the index of a symmetric operator). Moreover, guided by this theory, one can recover topology of sub-level sets of the form $M_\epsilon = f^{-1}((-\infty, \epsilon])$ as well. It is natural to apply it to the setting of machine learning,

taking $M$ to be the weight manifold and $f$ to be the training loss. In this case, $M_\epsilon$ are called loss landscapes and are of particular interest when $\epsilon$ is close to zero.

The most natural question to ask about $M_\epsilon$ from topological perspective is whether they are path-connected. This question has been considered in the deep learning literature before and experiments show that, indeed, it is often possible to connect two different weights of small loss by a path of almost constant small loss for a variety of different practically interesting architectures (see Raghavan and Thomson (2020),Garipov et al. (2018)). Connectedness turns out to be of practical importance because some points from the path normally have better generalization properties and better robustness properties against some types of adversarial attacks (see Izmailov et al., Zhao et al. (2019)). Later, some theoretical justifications for this phenomenon have been made as well for some classes of networks Kuditipudi et al. (2019). These justifications, however, are in the spirit of classical theoretical computer science rather than geometry or topology.

In the work presented here, we study mode connectivity via the lens of Morse theory. It can be perceived as building a connection between the saddle points and mode connectivity phenomena in a hope to shed more light on the structure of loss landscapes. Thinking from the perspective described above, one might become puzzled, as the following two facts seem to be contradictory to mode connectivity at first:

1) A common explanation for challenges underlying optimization with deep neural networks is the abundance of local minima.

2) According to Morse theory, a sub-level set $M_c$ cannot be connected if there is a local minimum with loss value $c$ different from the global minimum.

This multi-minima conjecture, however, is only in a seeming contradiction with the observation of mode connectivity. While parameter landscapes with many minima cannot, according to Morse theory, also be path connected as epsilon-loss level-sets, it is possible that their "thickened" versions are connected, where by "thickening" we mean letting the error increase to hit enough index one points to allow connectivity. If the "thickening" scenario takes place, this would mean that allowing some changes of loss is essential for the connectedness of the loss landscapes and is not just a drawback of using numerical methods for finding the low loss paths. Evaluating the losses at the corresponding index one points would clarify on how much of a loss change is required.

An alternative and simpler explanation could be that standard optimization algorithms actually converge to saddle points rather than genuine (local) minima for complex enough networks. We conducted experiments for a 2-MLP, a 3-MLP and a CNN architectures trained on MNIST and FashionMNIST tasks to find out which of the two possible explanations takes place.

Surprisingly for us, we observed the second scenario in all cases we considered: the loss landscapes contain saddle points that can be easily mistaken for local minima. Turns out that the saddle point convergence phenomenon has been observed before Dauphin et al. (2014). While this appears to explain mode connectivity in a lot of cases, we definitely would not completely rule out the first alternative as well. Thus, future directions include experimenting with bigger architectures and other optimizers to observe the first situation with required thickening in practise. In such cases it would be extremely interesting to identify and study the index one points of small loss responsible for connectivity.

## 2. Limitations

The experimental part of this work should be taken as preliminary. The networks (both MLP and CNN) that we studied empirically are small by modern standards and it is not immediately clear to us how to extend the same computations to much bigger networks. In addition, we tested the theory only on MNIST and FMNIST datasets. Finally, we used only two different optimizers: SGD and ADAM. As was noticed by an anonymous reviewer, using Hessian-aware optimizing techniques could be a real game-changer here.

## 3. Preliminaries

### 3.1. Morse Theory

We use the language of basic differential topology for this subsection. It is still accessible to people without familiarity with the latter, as the reader can just think of $M^n$ below as of a closed ball in $\mathbb{R}^n$. We will follow exposition from Milnor (2016) for a brief introduction to Morse Theory here.

Consider a smooth compact manifold $M^n$ equipped with a smooth function $f : M \to \mathbb{R}$. The following two definitions will be highly important to us.

- A point $p \in M$ is critical if $grad_p(f) = 0$.
- Given a map $\mathbb{R}^n \to U \subseteq M^n$, defining coordinates $x_1, \ldots, x_n$ around a point $x \in M^n$, the Hessian of $f$ at $x$ is $Hess_p(f) = (\frac{\partial^2 f}{\partial x_i \partial x_j})_{i,j}$. The operator $Hess_p(f)$ depends on the choice of coordinates in general, but the dimensions $n_+, n_0$ and $n_-$ of its positive, zero and negative eigenspaces respectively do not (according to Sylvester's law of inertia, which is a general fact about quadratic forms). Define index of $x$ as $ind(x) = n_-$, which is invariant to choice of coordinates as long as $f$ and $x$ are fixed.
- $f$ is said to be a Morse function if $Hess_p(f)$ is non-degenerate for any critical point $p$, i.e. if $n_0 = 0$.

Given a Morse function $f : M \to \mathbb{R}$, it is possible to reconstruct topology of $M$ from the knowledge of indexes of critical points of $f$ only. Moreover, defining $M_\epsilon = f^{-1}((-\infty, \epsilon])$, it turns out that it is also possible to reconstruct topology of $M_\epsilon$ knowing only indexes of critical points $p \in M_\epsilon$. We will not go into full detail about this reconstruction, but will cover enough for giving the necessary intuition. We need to introduce the following definition first:

- Given $0 \leq k \leq n$, we call $h_k = \mathbb{D}^k \times \mathbb{D}^{n-k}$ a $k$- handle. Of course, all these handles are the same topologically, that is up to a continuous deformation, but the gluing rules for reconstruction vary as we will see soon.

We assume that $M^n$ is compact and a basic theorem in Morse theory says that the set of critical points of a Morse function $f$ is always isolated (Corollary 2.3, p.8, Milnor (2016)). Thus, given the compactness assumption, this set is actually finite. Also due to the compactness, we know that $f$ is bounded from below and attains its minimum $m$. Thus, we know that $M_t = \emptyset$ for $t < m$ and starts changing for $m \geq t$. Now, assume that we know $M_t$ for $t < r$ and want to reconstruct $M_t$ further. Consider two cases:

- The pre-image $f^{-1}(r)$ does not contain any critical points of $f$. Since the set of critical points of $f$ is finite, it is possible to take a $\delta > 0$, such that there are no critical points in $f^{-1}((r - \delta, r + \delta))$ as well. Morse theory tells that in this case $M_{r+\delta}$ is diffeomorphic

to $M_{r-\delta}$ (Theorem 3.1, p.12, Milnor (2016)), so nothing changes if we do not hit a critical point.

- There is a critical $p$ with $f(p) = r$. Then $p$ is the only critical point satisfying this equation due to non-degeneracy of the Hessian. Again, take a $\delta > 0$, such that $p$ is the only critical point in $f^{-1}((r - \delta, r + \delta))$. According to Morse theory, $M_{r+\delta}$ can be obtained from $M_{r-\delta}$ by attaching $h_k$ to $M_{r-\delta}$. Moreover, this attachment is performed by gluing $\partial \mathbb{D}^k \times \mathbb{D}^{n-k} \subseteq \mathbb{D}^k \times \mathbb{D}^{n-k} = h_k$ to $\partial M_{r-\delta}$ (see the proof of Theorem 3.2 in Milnor (2016)).

**Remark 1** *If $k = 0$, $\partial \mathbb{D}^k = \emptyset$ and the gluing is just taking a disjoint union. The $k = 0$ case is the only case when the number of connected components can increase, as otherwise $\partial \mathbb{D}^k \times \mathbb{D}^{n-k}$ is non-empty and thus we just attach a connected set $h_k$ to a connected component of $M_{r-\delta}$.*

**Remark 2** *If $k = 1$, $\partial \mathbb{D}^k = \mathbb{S}^0$ is a disjoint union of two points and $\partial D^1 \times \mathbb{D}^{n-1} = \mathbb{D}^{n-1} \coprod \mathbb{D}^{n-1}$. Thus, attaching a 1-handle can glue two different connected components together. The $k = 1$ case is the only case when the number of connected components can decrease, as otherwise $\partial \mathbb{D}^k \times \mathbb{D}^{n-k}$ is connected, so it can touch only one connected component.*

### 3.2. Mode Connectivity

Mode connectivity is a recently introduced observation suggesting that, given two parameters $w_1$ and $w_2$ from the parameter space of a neural network, both giving very low loss, it is usually possible to connect them with a curve $\gamma(t), \gamma(0) = w_1, \gamma(1) = w_2$, such that $\gamma(t)$ is a low-loss parameter for any $0 < t < 1$ as well. That is, in words, that two low-loss weights can be connected by an entire small-loss path. One could naively guess that taking $\gamma(t) = tw_1 + (1 - t)w_2$ to be the straight line connecting the weights would do the job-note that this actually **far** from being the case: in fact, the accuracy can easily drop down to 20-30 percent for in-between points on the straight line even if the endpoints achieve 90+ percent accuracy Raghavan and Thomson (2020), Garipov et al. (2018). Thus, more sophisticated curves are needed. The main two practical approaches of finding the curves the authors are aware of are the following:

- Garipov et al. (2018) suggest using parametrized families of curves connecting $w_1$ and $w_2$, such as Bezier curves defined by $\gamma_\Theta(t) = (1 - t)^2 w_1 + 2\Theta t(1 - t) + 2tw_2$. Given such a parametrized family, the idea is to find the $\Theta$ minimizing the average loss $\int_0^1 L(\gamma_\theta(t))dt$ and use the corresponding $\gamma_\Theta$ as the desired low loss curve.

- Raghavan and Thomson (2020) suggest an iterative algorithm based on an approach that is also introduced in the same paper and that they call Functionally Invariant Paths.

One might wonder about the motivation behind mode connectivity studies. It can be justified in two possible ways:

- Conceptually, one of the mysteries of machine learning is that many different weight parameters can have the same train accuracy and it is unclear which setting of the parameters one should prefer. One of the natural ways of looking at it is introducing an equivalence relation on the set of parameters and calling two weights equivalent if the give approximately the same loss and if they can be connected by a curve of approximately the same loss. It is a very natural relation as it just means that one weight can be continuously

deformed into the other without changing the loss. In these terms, the mode connectivity comes out as surprising as it says that all low loss parameters are equivalent. Note, however, that this equivalence is measured only with respect to train loss and they still can be very different with respect to test loss as well as to other measures of generalization.

• From the applied point of view, as mentioned above, these paths can be found by efficient algorithms, and it is always helpful to have a lot of different candidates for a solution. Izmailov et al. use averaging along these paths to find solutions with better generalization performance. This resulted in the Stochastic Weight Averaging (SWA) algorithm, that has now been adopted as a PyTorch library. There are other noteworthy applications as well: for example, Zhao et al. (2019) and Raghavan and Thomson (2022) report that weights from the low loss paths tend to have better robustness properties with respect to some types of adversarial attacks.

Finally, we should mention here that there are some previous theoretical works on mode connectivity. Papers Kuditipudi et al. (2019) and Nguyen (2019) prove that it holds for certain types of networks, but they do not shed much light on what is going on from a geometric/topological perspective. Moreover, as was mentioned in the introduction and as will be elaborated in the next chapter in more detail, mode connectivity at first seems to contradict classical differential topology knowledge. There is also a recent work Horoi et al. (2022) that uses Topological Data Analysis to gain topological information about Loss Surfaces. This work, however, is mostly empirical and just uses TDA as a computational tool.

## 4. Morse Theory of Loss Landscapes

To apply Morse theory, one needs to make an appropriate choice of a smooth manifold $M$ and a smooth function $f : M \to \mathbb{R}$. Since we aim at studying Loss Surfaces, a reasonable choice would be taking $M = \mathbb{R}^D$, where $D$ is the dimension of the parameter space and $f(\Theta)$ to be the total loss of a fixed neural network with respect to a fixed dataset at the parameter $\Theta$. This, however, could raise two concerns: the manifold $M$ is usually assumed to be compact in the context of Morse theory and $f$ needs to have non-degenrate Hessians at all critical points. To address both, we suggest taking $f(\Theta) = L(\Theta) + \lambda ||\Theta||_2^2$ the sum of the train loss at the point $\Theta$ and an $l_2$-regularization term, where regularization term $\lambda$ should be "a very small number", in order to not affect the dynamics of SGD much. This way, we can take $M$ to be a closed ball of radius $O(\frac{1}{\lambda})$, as $f$ is very big outside of such a ball anyway, and it is not hard to make $f$ a Morse function this way, because $Hess(f) = Hess(L) + \lambda Id$, and one thus just needs to pick a $\lambda$ such that $-\lambda$ is not in the spectrum of the Hessians of critical points. We acknowledge, however, that this reasoning might come out a bit to loose, so one possible direction for future work could be making this part more rigorous. So, for the purposes of the present work, $M$ is a closed ball of a large radius (proportional to $\frac{1}{\lambda}$) and $f$ is the train loss with a very slight $l_2$-regularization added. Note that in this case we have a canonical coordinate system coming from the ambient Euclidean space and all the Hessian are calculated with respect to that coordinate system.

In these terms, mode connectivity tells us that $M_\epsilon$ is a connected set for small values of $\epsilon > 0$. To keep translating from the language of Morse theory to the language of optimization, a critical point of index $k$ is a saddle point with $k$ negative directions and $D - k$

positive. In particular, since the Hessian is positive-definite at any local minimum as can be easily seen from the corresponding Fourier approximation, local minima are always of index 0. Conventional wisdom of deep learning practitioners suggests that there usually are a lot of different local minima in the small loss valley. This comes out as puzzling because, if we take $\epsilon$ to be very close to the train loss at any of these minima, which does not happen to be the global minimum, the corresponding $M_\epsilon$ can be obtained from $M_{\epsilon'}$ with a close enough $\epsilon' < \epsilon$ by taking the disjoint union with $\mathbb{D}^D$(cf. Remark 1). In particular, such $M_\epsilon$ **cannot be connected**, directly contradicting the first sentence of this paragraph. Thus, a more profound analysis is needed to resolve this apparent paradox.

Going over the reasoning above, one might come up with two possible resolutions: either the points found by the optimization algorithm are usually saddle points rather than genuine local minima or, alternatively, it could be that these are genuine local minima, but there always are saddle points of index 1 very close to glue different components together (see Remark 2). We observed only the former in the experiments we ran: the points found usually turn out to have high indexes (see the next section for details). Moreover, turns out that this phenomenon has also been addressed in the literature Dauphin et al. (2014). While this appears to explain mode connectivity in a lot of cases, we definitely would not completely rule out the other aforementioned theoretical resolution as well, as it could be that to observe this scenario taking place in practise one has to experiment with significantly bigger networks or use different optimization methods. In any case, the present work should be perceived as a bridge between two seemingly non-related phenomena of mode connectivity and abundance of saddle points.

**Example 1**

*Let us consider the case of the linear regression as an illustration. Denote the data matrix by $X$ and the vector of labels by $y$. In this case, the train loss is $L(w) = ||Xw - y||_2^2$. Thus, it is easy to see that the corresponding Hessian is $Hess_w(L) = X^T X$ and is independent of $w$. This is non-negatively defined, but might be degenerate if the rows of $X$ are not linearly independent. If one adds a regularization term and considers $f(w) = ||Xw - y||_2^2 + \lambda ||w||_2^2$ instead, then one obtains $Hess_w(f) = X^T X + \lambda Id$, which is strictly positive for any $\lambda > 0$ no matter how small it is. In terms of optimization, this means that the corresponding problem is strictly convex. Morse theory tells us that the loss-landscape is diffeomorphic to the disc $\mathbb{D}^D$ in this case, which is perfectly consistent with convexity of the problem.*

## 5. Experiments

Numerical computations of indexes at critical points identified by SGD and ADAM could allow estimation of indexes and insights into architecture of these points as saddle points or true minima. As explained above, this is important for understanding mode connectivity in terms of Morse theory. As the index at a critical point is determined by the spectrum of the Hessian at this point by definition, we decided to plot the corresponding eigenvalue distributions at some of the critical points as well. It is interesting that the index seemed to be fairly similar across the critical points for each network and dataset, but the typical index value varies over different networks and datasets. It would be interesting to understand what determines this.

### 5.1. Set up

We considered the following architectures:

• A 2-layer MLP with the hidden layer of size 10. This gives 7960 parameters in total for images of size $28 \times 28$.

• A 3-layer MLP with two hidden layers of size 10. This gives 8070 parameters in total for images of size $28 \times 28$.

• A CNN architecture with two convolutional layers: both with 1 channel, kernel of size 5, stride 1, padding 2 and Maxpool kernel of size 2 and one linear layer mapping from $\mathbb{R}^{7 \times 7}$ to $\mathbb{R}^{10}$. This gives 552 parameters in total for images of size $28 \times 28$.

We trained the corresponding networks on MNIST and FMNIST datasets using Stochastic Gradient Descent with Cross-Entropy loss for the MLP and ADAM optimizer with the same loss for the CNN. After training, we calculated Hessians of the loss function with respect to the parameters. We then found the eigenvalues of the Hessians and calculated their indexes. The reader can see the corresponding plots in the next subsection.

### 5.2. Plotting indexes and eigenvalues

Recall that the index of a symmetric matrix is just the number of negative eigenvalues the matrix has. We found the index of the Hessian at a final point (that is at a point found by the optimizer) to be a fairly robust quantity under fixed architecture, dataset and optimization procedure. The optimizers did not converge to a single genuine minimum. In fact, the computations suggest that SGD and ADAM have discovered series of deep saddle points. Below we provide means and standard deviations of indexes for 10 runs and distributions of the eigenvalues of the Hessians for one of these runs. Our distributions are qualitatively similar to those reported in Papyan (2018) for different convolutional networks: they have large bulks around zero and some rare eigenvalues with spectral gaps on both sides of these bulks.

#### 5.2.1. CNN

We found the index to be $232.9 \pm 2.88$ for MNIST and $238.5 \pm 4.88$ for FashionMNIST.

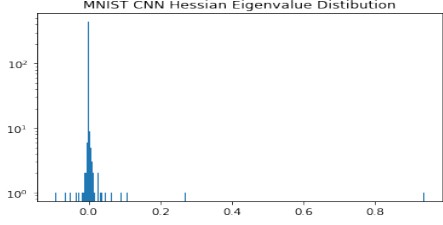
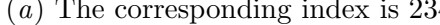
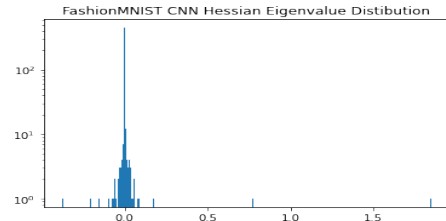

$(a)$ The corresponding index is 234     $(b)$ The corresponding index is 241

#### 5.2.2. 2-MLP

We found the index to be $1980.1 \pm 130.4$ for MNIST and $3637.9 \pm 119.92$ for FashionMNIST.

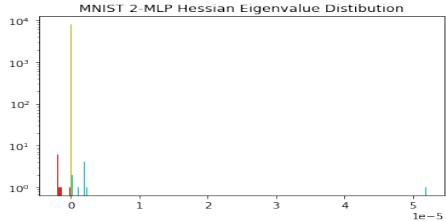
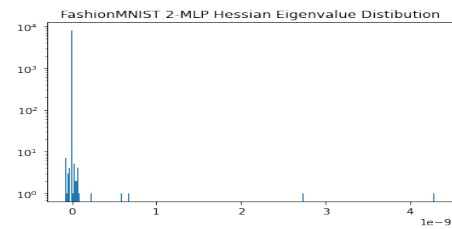

($c$) The corresponding index is 1939          ($d$) The corresponding index is 3712

### 5.2.3. 3-MLP

We found the index to be 2033.8±124.00 for MNIST and 3593.3±208.61 for FashionMNIST.

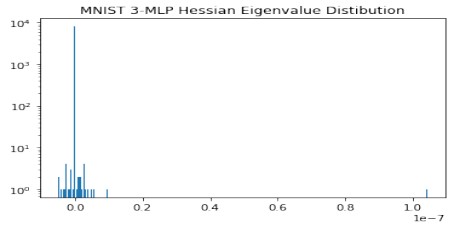
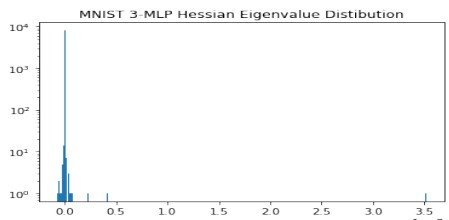

($e$) The corresponding index is 2125          ($f$) The corresponding index is 3688

## 5.3. We do hit essentially different solutions

Given that indexes of final points tend to take similar values under different runs of SGD and ADAM, one might be concerned that for some reason the optimizer just always hits a neighbourhood of one saddle point with near-optimal loss in this setting. To rule this out, we found final parameters for 500 initializations of the ADAM optimizer for the CNN and projected them onto $2D$ by applying PCA. We used a color map encoding the loss function: the darker the point is, the smaller the corresponding train loss is. The picture hints that there are multiple centers of clusters- black points, that are fairly far from each other in most cases, and the points with lighter colors were supposed to converge to the nearest black ones. One might be concerned here that reducing the situation to the 2D case might be too much of a simplification. Note, however, that if two points are observed far apart after being projected on a plane then it implies that they were even further apart before the projection.

## 5.4. Sanity check: linear regression case

Just as a sanity check, we prepared a random linearly separable dataset with 100 points and 2 classes in $\mathbb{R}^2$ and used linear regression with SGD training to separate them. Everything went as expected (see Example 1): the Hessian was the same at every point, it had index 0 and SGD would always converge to the same point for a big enough learning rate.

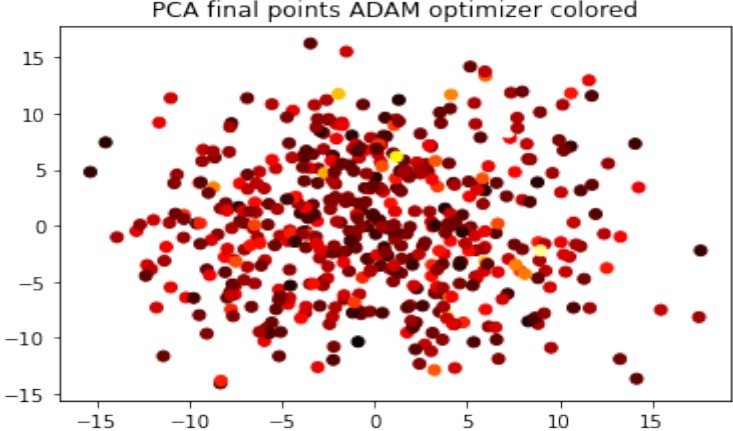

### 5.5. Indexes of random parameters

We also computed indexes of Hessians at parameters initialized randomly by PyTorch. Surprisingly, they turn out to be lower than indexes of Hessians of trained networks on average. This might indicate that neural network optimizers have a preference for finding high-index saddle points. The indexes for random parameters were found to be:

- 2-MLP: $1186.2 \pm 198.25$ for MNIST and $2840.9 \pm 387.35$ for FashionMNIST.
- 3-MLP: $1410.3 \pm 235.31$ for MNIST and $2721.9 \pm 272.12$ for FashionMNIST.
- CNN: $182.91 \pm 80.17$ for MNIST and $189.48 \pm 79.85$ for FashionMNIST.

### Acknowledgments

The authors would like to thank Guruprasad Raghavan for stimulating discussions as well as the anonymous referees for helpful suggestions that led to the improvement of the present paper. We acknowledge funding through the Packard Foundation and the Okawa Foundation.

## 6. Conclusion and future directions

We establish a connection between indexes of the Hessians of neural networks and topology of loss landscapes using Morse theory. We conclude in the cases we observe that mode connectivity arises because points found by optimizers are actually saddle points of big indexes rather than genuine local minima. This means that, even though these saddle points give small train losses, the train losses given by the actual local minima from the low loss valley must be yet smaller. Moreover, there must be an abundance of saddle points of index one giving smaller train losses than the points we converge to as well: they are necessary to glue different connected components corresponding to the different local minima together. In this sense, abundance of saddle points is a good thing, as otherwise mode connectivity would not be possible according to Morse theory. The observation that random networks have lower indexes on average than the final parameters found by the optimizers also suggests that saddle points might be a good thing after all.

A very interesting challenge for future work would be developing methods for decreasing the index without increasing the loss. A theoretical scheme for finding third-order local minima is suggested in Anandkumar and Ge (2016) and is proven to always work and to have polynomial time complexity. However, Anandkumar and Ge do not provide any experimental results and it was not clear to us whether it could be done practically and also it is unclear whether these third-order local minima would be of reasonably small loss. Nevertheless, trying to implement it and checking the results could be a good possible direction. Alternatively, as suggested by one of the anonymous reviewers, one could use other Hessian-aware optimization techniques, such as BFGS or LBFGS optimization methods, to shed more light on the structure of the loss landscapes.

Finally, we were intrigued by the spectral gaps that Hessians of loss functions of neural networks have. It goes beyond the scope of the present work, but it definitely deserves being studied on its own.

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
