# OpenReview forum: "Connectedness of loss landscapes via the lens of Morse theory"
_NeurIPS.cc/2022/Workshop/NeurReps — NeurReps 2022 Poster_

### Official Review · Reviewer_ZTRs · 2022-10-12
**Interesting work, but many important aspects left unaddressed**

**Confidence:** 5
**Soundness:** 2
**Presentation:** 2
**Contribution:** 3
**Overall Rating:** 3

**Summary:**

This paper analyses the loss landscape of neural networks by means of
Morse theory. Specifically, the paper aims to answer why the phenomenon
of mode connectivity, i.e. the existence of low-loss paths in the weight
space, is so prevalent in modern neural networks. To this end, the paper
models the weight set of a neural network as a point cloud, using an
extended loss function (consisting of the loss itself plus some
regularisation term) to obtain a Morse function. The extrema of said
Morse function may then be analysed.


**Questions:**

1. How is the underlying manifold $M$ created in practice? How are the
   weights "unrolled"?

2. What are additional references for the phenomenon discussed before
   Example 1? The current write-up just states "references", but does
   not provide any citations.

3. How is Example 1 connected to the overall goal of the paper? In
   Example 1, $X$ consists of a matrix of data points, whereas for the
   remainder of the paper, one is dealing with *weights*. What is the
   connection here?


**Limitations:**

Limitations are not discussed. I would suggest to incorporate an
additional paragraph on the modelling choices and to what extent they
"enforce" a certain structure on the weight space.


**Recommended Decision:**

1: Reject

**Relevance:**

4: Highly relevant

**Strengths And Weaknesses:**

The main strength of the paper lies in the creative use of Morse theory
to address the phenomenon of mode connectivity. To the best of my
knowledge, this is one of the first approaches utilising Morse theory in
that manner. As such, I appreciate the background description, which
serves to make the paper more accessible.

However, the current write-up is lacking important details and I have
the impression that some of the hypotheses necessitate much more
evidence. Specifically, I see the following issues:

- Important details on *how* all calculations are being done are missing
  from the write-up. If the full set of weights of a neural network is
  chosen to represent the underlying space, for instance, how is the
  training process being accounted for? Existing work considers the
  weights to form a trajectory in said space; is a similar perspective
  adopted here? Moreover, how are equivalences in the weight space being
  accounted for? The ordering of weights does not matter (from
  a puristic point of view), but for turning a neural network into
  a point cloud, some modelling decisions are required. These modelling
  decisions need to be elucidated in more detail.

- In addition, some aspects of the write-up are unclear to me. To what
  extent are the observed phenomena just an artefact of the choice of
  weight space model? As an additional test, I would suggest that paths
  in the extracted landscape are analysed, similar to the work by
  [Garipov et al. (Loss Surfaces, Mode Connectivity, and Fast Ensembling of DNNs)](https://proceedings.neurips.cc/paper/2018/file/be3087e74e9100d4bc4c6268cdbe8456-Paper.pdf).

- I don't understand the point on connectivity (the contradiction
  outlined in the introduction): in a Morse--Smale complex, local minima
  create basins of attraction, and I understand that any paths moving
  out from that region will have, by necessity, cross a different
  minimum. However, mode connectivity is explicitly defined by
  near-constant loss. To quote from the Garipov et al. paper above:

  > We show that the optima of these complex loss functions are in fact
  > connected by simple curves over which training and test accuracy are
  > nearly constant.

  Hence, the modes could still be connected in "thickened" sublevel
  sets? I am not sure whether I understand the contradiction correctly
  here.

- The analysis in Section 4.3 strikes me as preliminary. Going from
  a high-dimensional weight space to 2D surely cannot maintain
  a sufficient amount of structure?

I want to stress that I believe that this paper has the potential to
become a strong contribution to the field! While I cannot endorse it for
publication at the moment, I hope that my suggestions help in improving
the paper; I am very excited about the overall direction of addressing
loss landscape phenomena through the lens of Morse theory!

## Suggestions for improvement

- The indices of saddle points could be shown in Tables

- Figures on p. 7 could be relegated to the appendix and should be
  furnished with more explanations as well as a stand-alone caption.



**Submission Track:**

Proceedings Paper (9 Page)

---

> ### Author Response · Authors · 2022-11-01
> **Reviewer ZTRs response**
>
> First, we are grateful to the reviewer for taking the time to evaluate our work.
> We would like to apologize if the exposition was not clear enough. We do not treat neural networks as point clouds but rather just as vectors in the parameter space. As for analysis in section 4.3, the logic goes this way: if two points are far apart after a projection, then they were far apart before as well.  As for Example 1, the matrix of data points emerges after differentiating the error function by weights. We completely agree with other points made in the review and tried doing our best at incorporating these in the camera ready version as well as improving our exposition of the other points which the reviewer did not find clear.

---

### Official Review · Reviewer_q71x · 2022-10-14
**Great take on a metric for non convex optimization**

**Confidence:** 4
**Soundness:** 3
**Presentation:** 3
**Contribution:** 3
**Overall Rating:** 7

**Summary:**

After a very clear introduction to Morse theory, the authors present a way of understanding the topology and connectedness of loss landscapes via indexes of Hessian matrices.

They also present various experiments on classical machine learning datasets and neural network architectures (CNN and MLP).

The authors plot eigenvalues and corresponding index for various combinations of  dataset and neural network architectures.

**Questions:**

- Is the code available?
- Are there connexions/inspiration to be taken here with BFGS (or LBFGS) optimizers since they estimate the Hessian (or Approximate Hessian) matrix in order to solve the optimization problem?

**Limitations:**

- It was unclear to me how the method scales as the datasets and network get bigger. How would this method scale for an ImageNet dataset and resnet152 for instance?
- The first paragraph of the conclusion could be a little more clear on why saddle points "might" be a good thing. The conclusion seems to point that it is a good thing, but if the adjective might is employed this hints that it could not be good. Lots of literature has shown that saddle points are not that much of an issue when using SGD because the noise generated by the optimizer usually ends up pushing the optimizer out of the saddle point.

**Recommended Decision:**

3: Accept

**Relevance:**

3: Solid fit

**Strengths And Weaknesses:**

## Strenghts

- Taking the insight from Morse theory and looking at the indexes of the Hessians in order to get a better understanding of the loss landscapes is a very useful and interesting direction. The authors also explored the behavior of various datasets with regard to these indexes
- The authors present a very solid and clear background on morse Theory, mode connectivity, and morse theory of loss landscapes.
- Really appreciated that the authors sanity-checked their results with linear regression and made sure that the hessian was the same at every point.

## Weaknesses

- While the problem is tackled from the lens of morse theory it would have been interesting to see if any other approaches in the field of convex or non convex optimization would have been useful for a better understanding of the problem at hand. Maybe a related works section would have helped.

**Submission Track:**

Proceedings Paper (9 Page)

---

> ### Author Response · Authors · 2022-11-01
> **Reviewer q71x response**
>
> We would like to thank the reviewer for taking the time to evaluate our work. We address the expressed concerns and suggestions by adding a limitations section in the camera ready version.

---

### Official Review · Reviewer_Tsbr · 2022-10-15
**Using Morse theory is interesting, but the evaluation is quite superficial**

**Confidence:** 5
**Soundness:** 2
**Presentation:** 2
**Contribution:** 2
**Overall Rating:** 3

**Summary:**

The paper is trying to using Morse theory to study connectedness of loss landscape using Morse theory.

**Questions:**

It is unclear that why the topological information can detect that two local minimizers are connected by a part of almost constant small lost. This much be related to geometry as well.

In the set, the loss surface is essential a graph of the loss function defined all a Euclidean space (all parameters). If loss function is continuous, is the topology always trivial?

**Limitations:**

It is quite unclear to me why Morse index can immediately tell that two local minimizers are connected by a part of almost constant small lost. Experiment part are very weak.

**Recommended Decision:**

1: Reject

**Relevance:**

3: Solid fit

**Strengths And Weaknesses:**

Stelnghths: Using Morse theory to understand connectedness of Loss landscape could be useful.

Weakness: However, the major part of paper is just summarizing existing theory in textbook. The real connection to loss landscape part is relative weak. The experimental evaluation is far more than enough

**Submission Track:**

Proceedings Paper (9 Page)

---

> ### Author Response · Authors · 2022-11-01
> **Reviewer Tsbr response**
>
> We would like to start with thanking the reviewer for taking the time to evaluate our work.
> However, we do not see why the topology would necessarily have to be trivial for an arbitrary continuous function.
> It is always the case for convex functions, but does not appear to be the case for non-convex ones, which makes non-convex optimization interesting from the topological perspective as well. Also, the geometry of the Euclidean space can be taken into account to some degree by adding a regularization term to the error function.

---

### Decision · Program_Chairs · 2022-10-21

Accept (Poster)